# Cerebral Protection Strategies and Stroke in Surgery for Acute Type A Aortic Dissection

**DOI:** 10.3390/jcm12062271

**Published:** 2023-03-15

**Authors:** Leonard Pitts, Markus Kofler, Matteo Montagner, Roland Heck, Jasper Iske, Semih Buz, Stephan Dominik Kurz, Christoph Starck, Volkmar Falk, Jörg Kempfert

**Affiliations:** 1Department of Cardiothoracic and Vascular Surgery, Deutsches Herzzentrum der Charité (DHZC), 13353 Berlin, Germany; 2Charité—Universitätsmedizin Berlin, Corporate Member of Freie Universität Berlin, Humboldt-Universität zu Berlin, and Berlin Institute of Health, 10117 Berlin, Germany; 3DZHK (German Centre for Cardiovascular Research), Partner Site Berlin, 10785 Berlin, Germany; 4Department of Health Sciences and Technology, ETH Zurich, 8092 Zurich, Switzerland

**Keywords:** acute type A aortic dissection, cerebral protection, stroke, hypothermic circulatory arrest, cerebral perfusion, antegrade, retrograde

## Abstract

Background: Perioperative stroke remains a devastating complication in the operative treatment of acute type A aortic dissection. To reduce the risk of perioperative stroke, different perfusion techniques can be applied. A consensus on the preferred cerebral protection strategy does not exist. Methods: To provide an overview about the different cerebral protection strategies, literature research on Medline/PubMed was performed. All available original articles reporting on cerebral protection in surgery for acute type A aortic dissection and neurologic outcomes since 2010 were included. Results: Antegrade and retrograde cerebral perfusion may provide similar neurological outcomes while outperforming deep hypothermic circulatory arrest. The choice of arterial cannulation site and chosen level of hypothermia are influencing factors for perioperative stroke. Conclusions: Deep hypothermic circulatory arrest is not recommended as the sole cerebral protection technique. Antegrade and retrograde cerebral perfusion are today’s standard to provide cerebral protection during aortic surgery. Bilateral antegrade cerebral perfusion potentially leads to superior outcomes during prolonged circulatory arrest times between 30 and 50 min. Arterial cannulation sites with antegrade perfusion (axillary, central or carotid artery) in combination with moderate hypothermia seem to be advantageous. Every concept should be complemented by adequate intraoperative neuromonitoring.

## 1. Introduction

Although acute type A aortic dissection (ATAAD) is rare, it is a possibly lethal event that presents cardiac surgeons with major challenges [1,2,3]. Even though ATAAD is still associated with high mortality and morbidity rates, thirty-day mortality has dropped to an average of 17% [4]. Instant surgical treatment remains the therapy of choice and should be performed irrespectively of time or day [5,6,7,8]. This includes the resection of the entry tear in combination with an open distal anastomosis to prevent aortic rupture, re-establish antegrade true lumen perfusion and resolve malperfusion [9]. Perioperative stroke represents a devastating complication and is influenced by multiple patient specific factors [10,11,12]. Female patients, particularly, may suffer from an impaired neurological outcome [13]. Compared to elective aortic arch surgery, the incidence of stroke is significantly higher in surgery for ATAAD and may adversely affect post-operative outcomes [14]. Preoperative neurological dysfunction and cerebral malperfusion frequently appear in the setting of ATAAD and are associated with an increased risk of perioperative stroke [12,15]. To prevent this severe complication, different cerebral protection strategies evolved in the past. While systemic hypothermia prolongs the tolerance for cerebral ischemia, the use of a selective cerebral perfusion technique aims to meet the metabolic and oxygen demand of the brain in a direct matter. The aim of this narrative review is to provide an overview about cerebral protection strategies in ATAAD, discussing the most relevant topics in terms of cerebral protection based on the current scientific literature and give concrete clinical advice about how modern cerebral protection should be performed in surgery for ATAAD.

## 2. Materials and Methods

For this narrative review, the literature research was performed on Medline/PubMed using the following medical subject heading terms: “Cerebral Protection OR Cerebral Perfusion AND Acute Type A Aortic Dissection”. Only original articles dealing with cerebral protection strategies and neurological outcomes in the treatment of ATAAD published in 2010 or later were considered for further investigation. Each topic had to show potential impact on neurological outcome in the current literature and is, furthermore, of relevant clinical interest, playing a concrete role in the perioperative and surgical management of ATAAD. Studies reporting about cerebral protection strategies including patient cohorts that were not applicable (e.g., elective aortic surgery) or lacked detailed information regarding the different cerebral protection techniques, were excluded. The research process is illustrated in Figure 1. This review is a narrative review. We do not claim the conduction of a fully systematic review according to the PRISMA guidelines.

## 3. Results

In total, 24 original articles were identified for further investigation. The selected articles were stratified according to the recruited cerebral protection technique and examined for the presence of preoperative cerebral malperfusion, the time of circulatory arrest, the level of hypothermia and the rate of perioperative strokes. The occurrence of perioperative stroke was furthermore subdivided according to the most common neurological endpoints monitored in studies investigating cerebral protection techniques for ATAAD. Thus, postoperative neurological deficits were defined as permanent neurological deficits (e.g., stroke) **and** transient neurological deficits (e.g., transient ischemic attack, delirium etc.) **or** solely new postoperative neurological deficits (e.g., stroke), which were defined through the absence of preoperative neurological dysfunction. Furthermore, influencing factors identified by regression analyses—if performed—were added. The complete results are shown in Appendix A in Table A1. The gathered information is embedded in the following chapters, dealing with the most relevant aspects of neuroprotection in surgery for ATTAD with a specific focus on cerebral perfusion techniques.

### 3.1. Deep Hypothermic Circulatory Arrest

The use of deep hypothermic circulatory arrest (DHCA) was the first standardized approach of cerebral protection in aortic arch surgery. While the proximal anastomosis is made, the systemic temperature is usually lowered to between 16 and 19 °C. The use of DHCA offers the advantage to perform the open distal anastomosis in a bloodless and canula-free field without the use of clamps. Although this method leads to acceptable short-term results when a short circulatory arrest is needed, the risk for perioperative stroke and mortality may rise when circulatory arrest exceeds between 30 and 40 min [16,17,18]. Furthermore, Czerny et al. identified the use of isolated DHCA as an independent risk factor for permanent neurologic injury [19]. Additionally, the use of DHCA showed the poorest long-term survival compared to cerebral perfusion techniques according to Wiedemann et al. [20]. Based on these findings, isolated DHCA without selective cerebral perfusion is no longer recommended for cerebral protection in the operative treatment of ATAAD [9,21,22]. Especially in terms of more extensive, complex and time-consuming arch operations, the use of a selective cerebral perfusion technique as an adjunct to systemic hypothermia seems of upmost importance.

### 3.2. Retrograde Cerebral Perfusion

The idea of retrograde cerebral perfusion (RCP) is to complement DHCA by supplying additional oxygen to the brain through the venous and capillary system, thereby prolonging the tolerance of cerebral ischemia while flushing embolic material out of the brain vessels in a retrograde manner. For this purpose, a cannula is inserted into the superior vena cava through which oxygenated blood is applied during circulatory arrest with a flow rate of approximately 500 mL/min and a pressure of around 25 mmHg. A schematic example of RCP is shown in Figure 2. The use of RCP became popular after 1990 and was associated with a lower mortality, morbidity and perioperative stroke rate compared to DHCA alone [23,24]. Additionally, RCP enabled a prolongation of the safe cerebral ischemic time up to 40 min [17,25,26]. Although the use of RCP delivered excellent clinical results, doubts were cast about RCP regarding the true benefit of actual brain perfusion. Several studies demonstrating the superiority of RCP had been based on comparisons with historical cohorts treated with DHCA, potentially leading to bias in favor of RCP [23]. Notably, in a prospective clinical trial, Bonser et al. demonstrated no clinical or metabolic advantage of using DHCA in combination with RCP compared to DHCA alone [27]. Moreover, the potential effect of retrograde washout also remained questionable and is still lacking conclusive evidence [28,29]. Some have even tried to combine the flush effect of RCP with antegrade cerebral perfusion (ACP), but without any significant clinical advantage proven [30]. However, according to a recent meta-analysis, the additional use of RCP to DHCA is associated with a lower risk for perioperative stroke compared to DHCA alone, supporting the concept of an additional neuroprotective effect [31]. These results were also confirmed by a multicenter analysis by Ghoreishi et al. [32]. Mechanistically, the additional use of RCP may result in more homogenous cooling effects as well as an enhanced clearance of neurotoxic substances released due to latent cerebral ischemia. Nevertheless, the risk for a perioperative stroke during RCP may rise when circulatory arrest exceeds 60 min, which casts doubt about the use of RCP for longer circulatory arrest times [33,34]. Indeed, the implementation of RCP as a primary cerebral perfusion strategy is often performed under DHCA, which is associated with longer cardiopulmonary bypass times as well as hypothermia-related side effects [35,36]. The performance of RCP under higher temperatures in the range of moderate hypothermia has not been broadly established, but may be a valid option in cases of shorter circulatory arrest times not exceeding 30 min [37].

### 3.3. Antegrade Cerebral Perfusion

During antegrade cerebral perfusion (ACP), the brain is perfused in a physiological antegrade manner through the supra-aortic vessels. This allows to perform aortic arch surgery and treatment of ATAAD safely under moderate hypothermia from 26 °C to 28 °C, or even higher temperatures in selected cases, which leads to a lower cardiopulmonary bypass time and less hypothermia-related side effects [36]. For ACP, a target flow rate of 10 mL/kg bodyweight is applied, and the pressure is usually kept from around 40–70 mmHg. The cannulation of the right axillary artery constitutes the basis of ACP. In case of unilateral antegrade cerebral perfusion (uACP), the brachiocephalic trunk, the left common carotid artery and the left subclavian artery are clamped proximally to maintain sufficient perfusion pressure and to prevent a steal phenomenon. Furthermore, uACP can be expanded to bilateral antegrade cerebral perfusion (bACP) by additionally cannulating the left common carotid artery. Both techniques are illustrated in Figure 3. Another possibility for bACP is the cannulation of both common carotid arteries, offering the advantage of continuous cerebral blood flow without interruption. To get an overview, we identified all available studies since 2010 that investigated the neurological outcomes in ATAAD in terms of the applied selective cerebral perfusion techniques. So far, the majority of the studies have failed to detect a significant difference between RCP and ACP [14,21,38,39,40,41,42,43,44]. Moreover, prospective randomized trials are scarce, often limited due to small patients cohorts and have thus far not shown any significant difference regarding the risk for a perioperative stroke [45]. These findings suggest that similar neurological outcomes may be achieved using either RCP or ACP, justifying both techniques as legitimate and reproducible selective cerebral perfusion strategies, as recently recommended in the expert consensus of the American Association for Thoracic Surgery [6]. Moreover, in a propensity score-matched analysis, Montagner et al. recently showed that all three cerebral perfusion strategies lead to similar results during open zone 0 arch anastomosis for the treatment of ATAAD in terms of neurologic outcomes [38]. This may allow for tailored approaches addressing specific patient needs in the surgical treatment of ATAAD and adapt the procedure to individual conditions, e.g., the ability to implement RCP if cannulation of the supra-aortic vessels is not feasible due to substantial atherosclerosis. However, considering the advantages of moderate instead of deep hypothermia as well as the potentially limited safety of RCP for longer circulatory arrest times, ACP has been established as the main selective cerebral perfusion strategy in European cardiac centers, providing a suitable approach in cases of prolonged circulatory arrest [46,47].

### 3.4. Unilateral or Bilateral?

Although the implementation of ACP is recommended by current guidelines, there is no consensus regarding the specific technique of ACP in the setting of ATAAD [9]. This decision is often made on the basis of institutional policies, expert consensus and the surgeon’s preference. Due to its simplicity, uACP represents an attractive strategy for many surgeons, especially in cases of shorter circulatory arrest [48]. In contrast, bACP is currently the most frequently performed technique in Europe [46]. Assuming that uACP ensures an adequate perfusion of the whole brain, it is supported by several studies, which demonstrated equal results in terms of risk for perioperative stroke comparing uACP to bACP [18,49,50,51,52,53,54]. Norton et al. advocated the use of uACP on the basis of a similar incidence of right and left hemispheric strokes while avoiding the additional manipulation of supra-aortic vessels for bACP [53]. However, these findings also showed that bACP may not be associated with a higher risk for perioperative left hemispheric strokes. Interestingly, in a recent multicenter trial, Piperata et al. observed a higher incidence of perioperative as well as left hemispheric strokes in case of bACP compared to uACP [55]. So far, this is the only study reporting inferior neurological outcomes following bACP, highlighting the demand for further investigation. Regarding the potential time-consuming complexity of bACP, Tong and colleagues reported equal cardiopulmonary bypass and circulatory arrest times in comparison with uACP [56]. Although the incidence of perioperative strokes was not significantly lower, bACP led to a reduction in permanent neurologic deficits by 50% when compared to uACP. Additionally, the performance of uACP presupposes an intact circle of Willis to ensure an adequate perfusion of the left hemisphere, which does not apply to all patients undergoing surgery for ATAAD [57]. In contrast, Urbanski et al. demonstrated that the circle of Willis’ individual anatomical conditions do not correlate with potential contralateral hypoperfusion and that excellent neurological results can be achieved via uACP in elective aortic arch surgery [58,59]. A major limitation of most studies supporting the use of uACP is the fact that they often include a disproportionally higher number of hemiarch replacements and shorter circulatory arrest times (<40 min) when compared to complex aortic arch surgery [53,59]. This discrepancy was also highlighted by Preventza et al., leading to potential bias and possibly underestimating the efficacy of bACP because of its preferred use in extensive arch surgery [49]. Therefore, they suggested bACP as the perfusion strategy of choice if the estimated circulatory arrest time should extend >30 min. Notably, more recent studies have supported these suggestions with Angleitner et al. demonstrating superior long-term survival for bACP if the circulatory arrest time exceeded ≥50 min [52]. Consistently, a meta-analysis of Angeloni et al. revealed that longer circulatory arrest times were associated with increased mortality only among patients treated with uACP but not bACP [60]. The safety of uACP for shorter circulatory arrest times and the need of bACP for extended circulatory arrest was also confirmed by current results of the UK National Adult Cardiac Surgical Audit [22]. In case of total arch replacement in the setting of ATAAD, Liu et al. could even observe a significantly reduced risk for perioperative stroke in the group treated by bACP compared to uACP [61]. However, a clear recommendation for the limited safety time of circulatory arrest from which bACP should be applied has not been reliably described yet. The most recent guidelines suggest a time interval between 30 and 50 min of circulatory arrest, for which bACP may be advantageous [9]. The idea of performing trilateral cerebral perfusion via additional cannulation of the left axillary artery in selected cases represents a rather experimental approach with no available clinical evidence [62].

### 3.5. Arterial Cannulation and Neuroprotection

Femoral arterial cannulation has been used widely for the safe implementation of cardiopulmonary bypass. In recent years, increasing evidence suggests that femoral cannulation and consecutive retrograde perfusion may be inferior to antegrade perfusion techniques in terms of mortality and the risk for perioperative stroke [9,63,64]. Cannulation of the right axillary artery, particularly, in the setting of ATAAD may be associated with lower risk for perioperative stroke while being suitable for antegrade cerebral perfusion by clamping the innominate artery [65]. This may also be possible and safe in cases of innominate artery dissection [66]. Other approaches such as the direct central cannulation of the dissected aorta or bilateral cannulation of the common carotid arteries are possible cannulation strategies, but are only used at a few centers [9,67,68]. Double arterial cannulation strategies combining axillary and femoral cannulation are gaining interest with regards to potentially preventing and treating malperfusion syndromes, especially in cases of true lumen collapse and consecutive hypoperfusion [69]. Whether these techniques could also lower the risk of perioperative stroke is still under investigation.

### 3.6. Hypothermia and Neuroprotection

Because DHCA was the first established cerebral protection technique for aortic arch surgery during circulatory arrest, subsequently developed selective cerebral perfusion strategies were initially also performed during deep hypothermia. Then, especially in terms of ACP, the setting of deep hypothermia slowly switched to moderate hypothermia and reached average temperatures of approximately 28 °C [70,71]. Leshnower et al. compared the use of deep hypothermia in combination with ACP to moderate hypothermia with ACP in the treatment of ATAAD, showing no benefit in terms of mortality or perioperative stroke in the group with deep hypothermia [72]. These findings underline that there is no significant effect of additional cerebral protection when performing ACP under lower temperatures, justifying the use of moderate or maybe even mild hypothermia in the treatment of ATAAD [73]. In support, a recent propensity-matched analysis of the International Registry of Acute Aortic Dissection showed satisfactory results under moderate hypothermia while avoiding extended cardio-pulmonary bypass times under deep hypothermia [74]. Although deeper levels of hypothermia do not seem to offer an additional significant effect of cerebral protection during ACP, the risk of ischemic damage to the spinal cord must be considered as having particular relevance during prolonged circulatory arrest under moderate hypothermia [75]. Indeed, Kamiya et al. showed that the incidence of postoperative paraplegia increased from 0 to 18% when circulatory arrest exceeded ≥60 min and only moderate instead of deep hypothermia was used [76]. Of note, different sites of perioperative body temperature assessment have been delineated to diverge from the actual cerebral temperature, which may affect study outcomes. While tympanic temperature measurements were tightly correlated with arterial blood monitoring, bladder and rectal measurements resulted in significantly divergent measurements [77]. The optimal location for cerebral temperature measurement thus remains to be defined, although a combination of two locations may increase reliability.

### 3.7. Monitoring for Neuroprotection

Using near infrared spectroscopy (NIRS) to constantly monitor brain tissue oxygen saturation seems to be indispensable during surgery for ATAAD and is, furthermore, the most frequently used neuromonitoring technique in Europe during aortic arch surgery [46]. According to the individual baseline levels of cerebral oxygen saturation, proportional alterations can be immediately detected. This offers the opportunity to identify the occurrence of a sudden cerebral malperfusion syndrome while providing positive feedbacks after the successful restoration of dissected supra-aortic vessels with recovered adequate cerebral perfusion. Furthermore, when applying uACP with asymmetric oxygen saturation reflected by decreases over the left hemisphere, NIRS can support the decision to switch to bACP, which, in turn, may lower the risk for developing a perioperative stroke [78]. Though NIRS represents an essential and useful tool during aortic arch surgery, its registration range is restricted to the frontal cortex when placed bilaterally on the forehead. Thus, local perioperative strokes in the frontal lobes or global oxygen declines in one or both hemispheres can be reliably detected, whereas strokes in the medial or vertebrobasilar area may not be recognized.

### 3.8. Neuroprotective Drugs

The perioperative application of drugs with potential neuroprotective effects, such as barbiturates, steroids or mannitol, is also frequently discussed in aortic arch surgery. An analysis of the German Registry for Acute Aortic Dissection Type A revealed that 50% of patients undergoing surgery for ATAAD received at least one neuroprotective drug, indicating a slight advantage for the use of steroids in terms of neurological outcomes [79]. However, further scientific evidence supporting the implementation of neuroprotective pharmacological treatment is low, while no significant benefit has been shown in larger clinical trials yet [80].

## 4. Conclusions

The chosen concept of cerebral protection in the surgical treatment of ATAAD has a great impact on patients’ outcome in terms of mortality and the risk for a perioperative stroke. Whereas the use of DHCA alone seems to be outdated today, RCP and ACP have both been established as legitimate and reproducible selective cerebral perfusion strategies. Thereby, the use of ACP seems to be superior regarding the possibility to perform the operation safely under moderate hypothermia compared to RCP and deep hypothermia. Although there is no clear recommendation by current guidelines for the technique of ACP, several studies suggest bACP as the most suitable cerebral perfusion strategy for prolonged circulatory arrest, while uACP is proven to be safe for shorter times of circulatory arrest. A well-described and evidence-based time threshold for this purpose is still missing, although a time interval from 30–50 min has recently been proposed. This issue demands a prospective, randomized and ideally multicenter trial to evaluate the true benefit of bACP. Arterial cannulation sites with consecutive antegrade body perfusion, especially right axillary cannulation, may lead to lower mortality and superior neurological outcome. Single arterial femoral cannulation leading to retrograde body perfusion should hereby be avoided, if possible. Regarding the level of hypothermia, moderate hypothermia at approximately 28 °C seems to be an effective and well-balanced strategy, offering adequate cerebral protection while avoiding the disadvantages of deep hypothermia. Additionally, precise temperature management and adequate neuromonitoring should complement the concept of cerebral protection. For this purpose, NIRS plays a major role in neuromonitoring during aortic arch surgery, possibly detecting and potentially lowering the risk for developing a perioperative stroke.

## Figures and Tables

**Figure 1 jcm-12-02271-f001:**
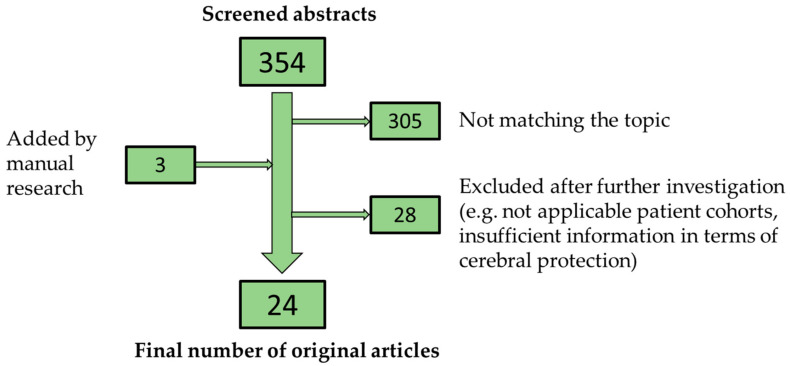
Process of the literature research.

**Figure 2 jcm-12-02271-f002:**
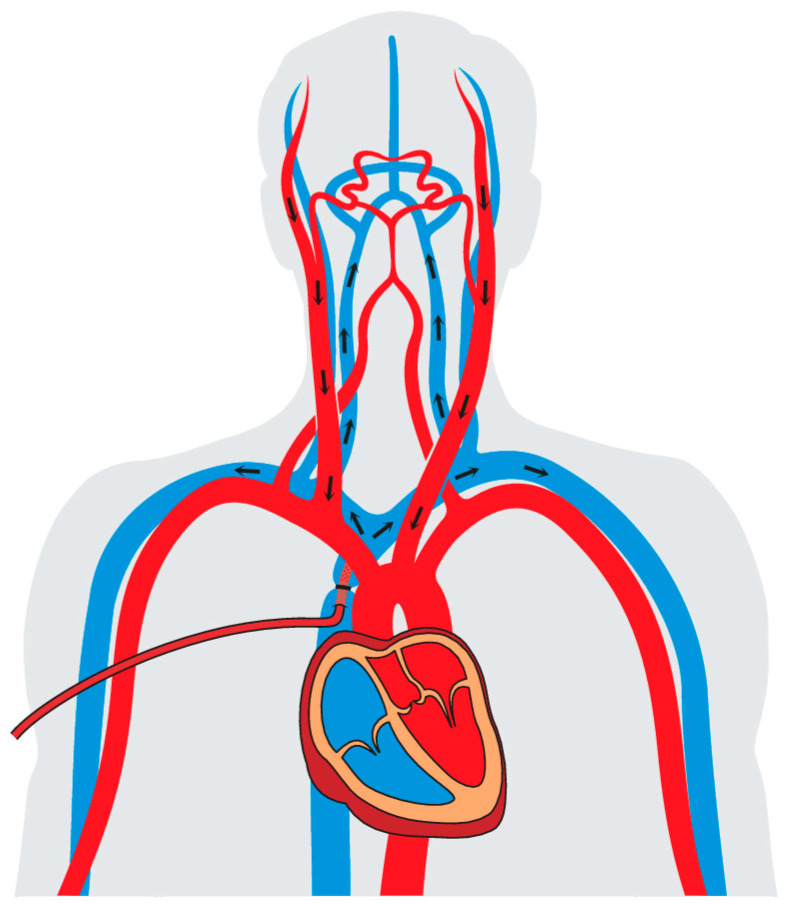
Performance of retrograde cerebral perfusion.

**Figure 3 jcm-12-02271-f003:**
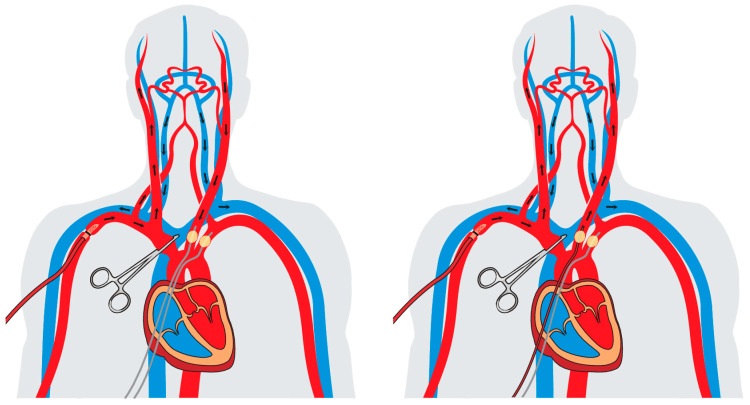
Performance of unilateral (**left**) and bilateral antegrade cerebral perfusion (**right**).

## Data Availability

Not applicable.

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
