# Peer review of "Cerebral Protection Strategies and Stroke in Surgery for Acute Type A Aortic Dissection"

_jcm, 2023, doi:10.3390/jcm12062271_

Round 1

Reviewer 1 Report

I would like to congratulate the authors for this well written and comprehensive review about this important topic. They summarized the most relevant publications that address the debatable cerebral perfusion strategies with sound recommendations based on the available published data.   

Author Response

Response:

Dear Reviewer 1,

thank you for your comment. We are pleased to get a chance to contribute on this important topic in the Journal of Clinical Medicine. Please don't hesitate to contact us for any further changes in the manuscript.

Kind regards on behalf of all co-authors,

Leonard Pitts, M.D.

Reviewer 2 Report

The methods are insufficient. Please follow prisma guidelines for a systematic review of the literature.

What was the aim of the review?

How were relevant topics chosen?

Author Response

Dear Reviewer 2,

thank you for your comments. We elaborated on every point as follows:

Point 1: The methods are insufficient. Please follow prisma guidelines for a systematic review of the literature.

Response 1: Thank you for your comment. We totally agree with your point. Indeed, following the PRISMA guidelines would have increased the methodological quality. Due to the multitude of influencing factors as well as the heterogeneity of studies in the field of ATAAD research, we decided to conduct a narrative review. This enabled a broad discussion about different topics which all have potential influence on the neurological outcome in surgical treatment of ATAAD. Therefore, we did not follow the PRISMA guidelines completely, taking into account that the methodological quality will be affected. We do not claim the conduction of a fully systematic review according to the PRISMA guidelines. To state this clear we added a paragraph to the methods section where we highlight this limitation for the reader. Changes can be seen in the re-uploaded version of our manuscript (page 2, line 67-68).

Point 2: What was the aim of the review?

Response 2: Thank you for your question. The aim of this narrative review is as follows:

  • Providing an introduction and overview of cerebral protection in surgery for ATAAD
  • Discussing the most relevant topics in terms of cerebral protection in surgery for ATAAD based on current scientific literature
  • Give relevant and concrete clinical advice how cerebral protection should be performed in surgery for ATAAD

To state this clear for the reader, we added a paragraph in the introduction part. Changes can be seen in the re-uploaded version of our manuscript (page 2, line 52-55).

Point 3: How were relevant topics chosen?

Response 3: Thank you for your question. The following topics were chosen for investigation and discussion in the context of cerebral protection in surgical treatment of ATAAD

  • Selective cerebral perfusion strategies
  • Circulatory arrest time
  • Hypothermia
  • Arterial cannulation strategies
  • Neuromonitoring
  • Neuroprotective drugs

These topics were chosen based on current guidelines (2022 ACC/AHA Guideline for the Diagnosis and Management of Aortic Disease) and on the results of clinical studies investigating the impact of cerebral protection strategies on the risk for perioperative stroke in surgical treatment of ATAAD. Each topic has been shown to have a potential impact on neurological outcome. Furthermore, each topic is of relevant clinical interest, playing a concrete role in the perioperative and surgical management of ATAAD. To state this clear for the reader, we added a paragraph in the methods section. Changes can be seen in the re-uploaded version of our manuscript (page 2, line 61-64).

Reviewer 3 Report

Dear Authors,

the Manuscript is very interesting and well organized. The Manuscript is about an important issue to keep in mind during type A aortic dissection treatment. It is very important, in fact, to have an effective perfusion technique, to reduce the risk of perioperative stroke. The Manuscript is a well written and clear Review, debating about the best cerebral protection strategies, listing the current methods described in scientific literature, and for each of them exposing their main advantages and/or disadvantages. It is easy to understand and it is a good overview of current literature about the topic.

Author Response

Dear Reviewer 3,

thank you for your comment. We are pleased about the positive feedback and appreciate to get a chance to contribute on this important topic in the Journal of Clinical Medicine. Indeed, our aim is to give an overview about relevant topics in terms of cerebral protection in surgical treatment of ATAAD as well as give concrete, evidence-based and clinically relevant advise how modern cerebral protection should be performed.

Please don't hesitate to contact us for any further changes in the manuscript.

Kind regards on behalf of all co-authors,

Leonard Pitts, M.D.

Reviewer 4 Report

Thank you for asking to review the manuscript entitled "Cerebral Protection Strategies and Stroke in Surgery for Acute Type A Aortic Dissection". My compliments to the authors for this thorough evaluation of the current literature. The review doesn’t really add much in terms of current knowledge, but it is a clear and comprehensive analysis in the matter of cerebral protection for the repair of acute aortic syndromes. It is definitely appropriate for scientific divulgation. 

Author Response

Dear Reviewer 4,

thank you for your comment. We appreciate the possibility to contribute on this important topic in the Journal of Clinical Medicine. Though we do not add new findings or data to the topic of cerebral protection in ATAAD, we aim to give an overview as well as give concrete, evidence-based and clinically relevant advise how modern cerebral protection should be performed. As a narrative review, our intention is to give a comprehensive and well readable essay for daily clinical practice.

Please don't hesitate to contact us for any further changes in the manuscript.

Kind regards on behalf of all co-authors,

Leonard Pitts, M.D.

Reviewer 5 Report

Overall, thorough and nicely written manuscript summarizing modern days technique for stoke prevention during Acute Type A Aortic Dissection surgeries.

Recommend addressing following comments:

Page 3 , lines 69-70: Recommend to include all disease/pathophysiology/deficits which were mentioned in the studies.

Page 3, line 115: Spelling error? " colling effects"?

Page 4: line 148: Spelling error? "vindicating"? 

Author Response

Dear Reviewer 5,

thank you for your comments. We elaborated on every point as follows:

Point 1: (Page 3 , lines 69-70) Recommend to include all disease/pathophysiology/deficits which were mentioned in the studies.

Response: 1 Thank you for your comment. In terms of neurological outcome we included all available endpoints for perioperative stroke. Those are mainly described as transient or permanent neurological deficits caused by perioperative stroke. Considering the patients' preoperative neurological condition, some studies reported the incidence for stroke as new perioperative stroke which is defined by the absence of preoperative neurological dysfunction in general. To ensure consistency, we decided to use these uniform definitions and make interpretations more clear. We totally agree with you that other deficits which carry a neurological burden (e.g. coma, spinal ischemia) as well as other primary endpoints (e.g. mortality) play also a major role in the surgical treatment of ATAAD. Focusing on the incidence of perioperative stroke and strategies for cerebral protection, we decided not to report other endpoints. Therefore, we did not present and discuss these data specifically. However, this ensured a concise design and comprehensible results, leading to our primary goal to give concrete advice and easy readable information to the reader how modern cerebral protection in surgery for ATAAD should be performed.

Point 2: (Page 3, line 115) Spelling error? " colling effects"?

Response 2: Thank you for your comment. We corrected the spelling error properly. Changes can be seen in the re-uploaded version of our manuscript (page 4, line 125).

Point 3: (Page 4: line 148) Spelling error? "vindicating"? 

Response 3: Thank you for your comment. We corrected the spelling error properly. Changes can be seen in the re-uploaded version of our manuscript (page 5, line 158).